# Maternal and Neonatal Factors Modulating Breast Milk Cytokines in the First Month of Lactation

**DOI:** 10.3390/antiox12050996

**Published:** 2023-04-25

**Authors:** David Ramiro-Cortijo, Gloria Herranz Carrillo, Pratibha Singh, Miguel Rebollo-Hernanz, Pilar Rodríguez-Rodríguez, Santiago Ruvira, María Martín-Trueba, Camilia R. Martin, Silvia M. Arribas

**Affiliations:** 1Department of Physiology, Faculty of Medicine, Universidad Autónoma de Madrid, 28029 Madrid, Spain; 2Division of Gastroenterology, Beth Israel Deaconess Medical Center, Harvard Medical School, Boston, MA 02215, USA; 3Instituto Universitario de Estudios de la Mujer (IUEM), Universidad Autónoma de Madrid, 28049 Madrid, Spain; 4Division of Neonatology, Instituto de Investigación Sanitaria del Hospital Clínico San Carlos (IdISSC), Hospital Clínico San Carlos, 28040 Madrid, Spain; 5Department of Agricultural Chemistry and Food Science, Institute of Food Science Research CIAL (UAM-CSIC), Universidad Autónoma de Madrid, 28049 Madrid, Spain; 6Department of Neonatology, Beth Israel Deaconess Medical Center, Harvard Medical School, Boston, MA 02215, USA; 7Division of Translational Research, Beth Israel Deaconess Medical Center, Harvard Medical School, Boston, MA 02215, USA

**Keywords:** cytokines, breast milk, prematurity, neonatal adverse outcome, sex, inflammatory diet, total antioxidant capacity, lipid peroxidation

## Abstract

Breast milk (BM) cytokines support and modulate infant immunity, being particularly relevant in premature neonates with adverse outcomes (NAO). This study aimed to examine, in a cohort of Spanish breastfeeding women, changes in BM cytokines in the first month of lactation, their modulation by neonatal factors (sex, gestational age, and NAO), maternal factors (obstetric complications, C-section, and diet), and their relationship with oxidative status. Sixty-three mother-neonate dyads were studied at days 7 and 28 of lactation. Dietary habits were assessed by a 72-h dietary recall, and the maternal dietary inflammatory index (mDII) was calculated. BM cytokines (IL-10, IL-13, IL-8, MCP-1, and TNFα) were assessed by ultra-sensitive chemiluminescence. Total antioxidant capacity was assessed by the ABTS method and lipid peroxidation by the MDA+HNE kit. From days 7 to 28 of lactation, the levels of IL-10 and TNFα remained stable, while IL-13 increased (β = 0.85 ± 0.12, *p* < 0.001) and IL-8 and MCP-1 levels decreased (β = −0.64 ± 0.27, *p* = 0.019; β = −0.98 ± 0.22, *p* < 0.001; respectively). Antioxidant capacity and lipid peroxidation also decrease during lactation. Neonatal sex did not influence any of the cytokines, but BM from mothers with male infants had a higher antioxidant capacity. Gestational age was associated with male sex and NAO, being inversely correlated with the BM proinflammatory cytokines IL-8, MCP-1, and TNFα. From days 7 to 28 of lactation, BM from women with NAO infants increased MCP-1 levels and had a larger drop in antioxidant capacity, with the opposite trend in lipid peroxidation. MCP-1 was also significantly higher in women undergoing C-section; this cytokine declined in women who decreased mDII during lactation, while IL-10 increased. Linear mixed regression models evidenced that the most important factors modulating BM cytokines were lactation period and gestational age. In conclusion, during the first month of lactation, BM cytokines shift towards an anti-inflammatory profile, influenced mainly by prematurity. BM MCP-1 is associated with maternal and neonatal inflammatory processes.

## 1. Introduction

Cytokines play an important role during intrauterine and extrauterine development and are also present in breast milk (BM), being essential for the normal development of the neonatal immune system, epithelial barrier function, and microbiome [1]. BM cytokines are derived from the epithelial cells of the mammary gland, from BM immune cells, or transferred from the maternal circulation [2,3]. Both anti- and pro-inflammatory cytokines are found in BM [3,4]. Anti-inflammatory cytokines, such as IL-10 and IL-13, promote B cell differentiation and immunoglobulin production, playing a regulatory role in the neonatal gastrointestinal tract [5,6]. Pro-inflammatory cytokines, such as IL-8, tumor necrosis factor-α (TNF-α), and some chemokines, can induce systemic inflammation [1]. However, they also have some beneficial effects. IL-8 is a chemoattractant for leukocyte recruitment from maternal circulation into the BM and plays a role in the defense against gastrointestinal and respiratory diseases [5,6]. Monocyte chemotactic protein 1 (MCP-1), a small heparin-binding chemokine secreted by monocytes, macrophages, and dendritic cells, also participates in the recruitment of immune cells to the site of infection [7], and TNFα can interact with the neonatal intestinal lumen [6], triggering mucin secretion in the gut and respiratory system [8].

BM cytokine profile changes during lactation can be modulated by both maternal and neonatal factors [9]. BM pro-inflammatory cytokines are higher in women with inflammatory conditions, such as allergies or celiac disease [10,11], and in women in stressful situations [1,12]. Exposure to pollutants also dysregulates BM cytokine balance towards a pro-inflammatory profile and can potentially contribute to the development of immune disorders later in life [13]. Preterm delivery can also modify the BM cytokine profile. This is particularly relevant since premature infants present several common morbidities associated with inflammation [14] and exhibit an underdeveloped immune system with low levels of cytokines in response to inflammatory insults [1,15,16]. However, the influence of preterm delivery on BM cytokine levels is still controversial. Some studies demonstrate higher levels of BM pro-inflammatory cytokines [17], while others show no significant effect of gestational age and suggest that maternal diet would be more relevant [5]. In fact, nutrition is an important modulator of inflammation, and it has been shown that the maternal dietary pattern regulates BM immune components [18], and a low-quality diet affects BM cytokine levels [19]. Dietary components that modulate inflammation, particularly long-chain polyunsaturated fatty acids (LCPUFAs), have been the focus of attention. However, data regarding their role is still inconsistent. Some studies show that maternal Western diets low in omega (n-3) LCPUFAs decrease the levels of BM anti-inflammatory cytokines [20], while others do not evidence an influence [21]. In this context, the use of a diet inflammatory index (DII) score may be useful [22]. This index has been used to evaluate the influence of nutrition on diseases related to inflammation, including subclinical mastitis in lactating women [23]. However, there is a lack of studies investigating the relationship between maternal DII and BM cytokines. Maternal inflammatory conditions during gestation and delivery, such as preeclampsia, gestational diabetes, or C-section, may also influence the BM cytokine profile. Furthermore, it is possible that the BM cytokine profile may be related to oxidative status, given the close relationship between inflammation and oxidative stress [24,25].

The aims of this study were to examine, in a cohort of Spanish breastfeeding women, changes in BM cytokine levels in the first month of lactation, evaluating the modulation by maternal and neonatal factors. We have also analyzed if there is an association between BM cytokine pattern and oxidative status.

## 2. Materials and Methods

### 2.1. Study Design and Cohort

In this observational, longitudinal, and non-interventional study, the women were enrolled within the first 72 h postpartum, from September 2019 to March 2020, at the Obstetrics and Gynecology and Neonatology Departments of Hospital Clínico San Carlos (HCSC, Madrid, Spain). Maternal inclusion criteria were: ≥18 years old, good understanding of the Spanish language, and undiagnosed disease at the time of study. Mothers with specific dietary restrictions, such as those following a diet for competition sports, vegetarians, and vegans, were excluded from the study. The mothers who agreed to participate in the study signed an informed consent form.

Since prematurity is associated with an underdeveloped immune system and neonatal adverse outcomes (NAO) with an inflammatory component, we were particularly interested in the assessment of preterm delivery as a possible factor affecting BM cytokines. Therefore, a sample size was calculated considering the rate of premature births in Spain is 6.3% [26], with a 5% margin of error and a statistical power of 80%. The estimated sample size was 46 women. However, to reduce the risk of bias in this observational study, we increased the sample size beyond the estimated number. Thus, the final cohort included 63 women. From them, 51 provided a BM sample at 7 ± 1 days, and 46 provided a sample at 28 ± 1 days postpartum. The flow chart with recruitment, follow-up, and withdrawals is shown in Figure 1.

The present study was performed following the Declaration of Helsinki for studies on human subjects and was approved by the Ethical Committee of the HCSC (Ref. 19/393-E). Additionally, the study follows the STROBE statement and checklist criteria for cohort design [27].

### 2.2. Social and Clinical Data

Close to enrollment day, the women filled out a sociodemographic questionnaire, including maternal age, educational level, employment situation, and nationality.

The following obstetrical and labor clinical data were collected from medical records: number of gestations, spontaneous pregnancy (yes/no), gestational anemia (hemoglobin levels < 11 g/dL), gestational diabetes (defined as a positive result in the 100 g oral glucose tolerance test), preeclampsia (blood pressure > 160/110 mmHg with proteinuria or thrombocytopenia after 20 weeks of gestation), C-section (yes/no), and gestational age (completed weeks of gestation). Women undergoing C-section had antibiotherapy during the first 24–48 h postpartum. Women who gave birth before 37 weeks of gestation were categorized as premature delivery, and those who gave birth after 37 weeks were categorized as full-term gestation.

The neonate parameters collected from medical records were sex, Apgar score at 5 min, antibiotherapy (yes/no), and presence of NAO. According to the neonatologist guidelines of the HCSC, NAO was considered if neonates were diagnosed with any of the following comorbidities: cardiac alterations (persistent patent arterial ductus or presence of hemodynamic alterations based on echocardiographic and clinical criteria); neurological alterations (cerebral white matter lesions, germinal or intraventricular hemorrhage (IVH) defined according to Volpe’s classification [28,29]); Chronic Neonatal Lung Disease (CNLD), defined as neonate requirement for supplemental oxygen during first 28 days of life and clinical lungs evaluation at ≥36 weeks of postmenstrual age; [30]); Necrotizing Enterocolitis (NEC), considered infant at stage IIA and above, according to a modified Bell’s criteria; [31]); Late Onset Sepsis (LOS), defined as clinical signs of septicemia along with a positive blood culture after 72 h).

### 2.3. Maternal 72-h Dietary Intake

Maternal dietary pattern was evaluated using the 72-h dietary recall (72hDR), which involved open-ended questions to obtain women’s food intake [32,33]. The 72hDR was collected at days 7 ± 1 and 28 ± 1 of lactation. The participant provided information about the ingredients, preparation methods, and quantities of all the food, including any supplements and water, consumed during the preceding three days. Women were instructed to use the Spanish version of “Visual Guide of Food and Portions” [34] to obtain detailed information.

In the records, the portion size of food was also recorded, and the resulting data were converted into daily units of macronutrients, micronutrients, vitamins, and minerals using the DIAL software (version 3.11.9, Alce Ingeniería, Madrid, Spain). The data provided information about: energy (Kcal), proteins (g), fats (g), saturated fatty acids (SFAs; g), monounsaturated fatty acids (MUFAs; g), polyunsaturated fatty acids (PUFAs; g), total cholesterol (Chol; mg), carbohydrates (g), fiber (g), vitamin A (retinols; μg), vitamin B1 (thiamin; mg), vitamin B2 (riboflavin; mg), vitamin B3 (niacin; mg), vitamin B6 (pyridoxines; mg), vitamin B9 (folic acids; μg), vitamin B12 (cobalamin; μg), vitamin C (ascorbic acid; mg), vitamin D (μg), vitamin E (α-tocopherols; mg), and iron (Fe; mg). The dietary variables were scaled by typification for the data analysis.

### 2.4. Calculation of the Maternal Dietary Inflammatory Index

The 72hDR was used to calculate a maternal dietary inflammatory index (mDII) based on the global composite data set from the Dietary Inflammatory Index Development Study of Shivappa et al., linking diet with plasma high-sensitivity C-reactive protein values [22]. Briefly, the macronutrients, micronutrients, minerals, and vitamins of 72hDR were standardized, then balanced by the overall inflammatory score of each parameter to estimate the nutrient-inflammatory effects for each individual food parameter. DII provides a theoretical list of 45 foods [22], and we used the 20 available foods from the 72hDR, as other authors have described [23,35]. The mDII was calculated as follows:mDII = 0.106 × B12 − 0.365 × B6 + 0.097 × Carbohydrates + 0.110 × Chol + 0.180 × Energy + 0.298 × Fats − 0.663 × Fiber − 0.190 × B9 + 0.032 × Fe − 0.009 × MUFAs − 0.246 × B3 + 0.021 × Proteins − 0.337 × PUFAs − 0.068 × B2 + 0.373 × SFAs − 0.098 × B1 − 0.401 × A − 0.424 × C − 0.446 × D − 0.419 × E

The mDII was categorized as low or high considering the median of the variable, using previously described methodology [23], with slight modifications. Women were also stratified in three clusters according to the change in mDII from day 7 to day 28 of lactation: (1) women that maintained the mDII category (low or high), (2) women who decreased the mDII category (high to low), and (3) women who increased the mDII category (low to high).

### 2.5. Breast Milk Collection and Processing to Obtain the Defatted Phase

One mL of BM was collected at 7 ± 2 and 28 ± 2 days of lactation by each woman using an electric breast pump (Symphony^®^ Medela, Barcelona, Spain). Women were instructed to wash their hands and clean their breasts with a gauze with soap and water before self-expressing milk and to obtain the milk always after neonate feeding. BM was collected between 10:00 a.m. and 11:59 a.m. from both breasts, pooled, transferred to a glass bottle, and stored in the freezer. The time between extraction and processing took a maximum of 3 h. The BM samples were then centrifuged three times (2000× rpm for 5 min at 4 °C) to obtain the defatted phase. Glass serological pipettes were used to extract the aqueous layer, placed in a clean tube, and stored at −80 °C until use.

### 2.6. Breast Milk Cytokine Detection

The levels of IL-10, IL-13, IL-8, MCP-1, and TNFα were measured in the defatted phase of BM using the Human U-Plex Ultra-Sensitive MSD Kit (Meso Scale Diagnostics, LCC, Rockville, MD, USA), following the manufacturer’s protocol. Briefly, each linker vial was conjugated with 200 μL of biotinylated antibody and incubated for 30 min at room temperature. Then, 200 μL of the stop solution was added to each linker vial and incubated for 30 min at room temperature. To prepare the multiplex coating solution, the linker-conjugated antibodies were mixed in a clean tube, obtaining a total volume of 6 mL. A 50 μL volume of the multiplex coating solution was added to each well. The plate was covered and placed on a shaker for 1 h at room temperature. Thereafter, the plate was washed three times with 150 μL of 0.05% Tween-20 PBS 1X solution (*v*/*v*).

The standard curve was prepared according to the manufacturer’s guidelines, and the BM defatted samples were diluted 1.2-fold. A 50 μL volume of the standard curve or the diluted samples was added to the plate in duplicate. Thereafter, the plate was incubated under shaking conditions for 2 h at room temperature. Then, the plate was washed three times with 150 μL of 0.05% Tween-20 PBS 1X solution (*v*/*v*). A 50 μL volume of the detection antibody was added to each well, and the plate was incubated under shaking conditions for 1 h at room temperature. Thereafter, the plate was washed three times with 150 μL of 0.05% Tween-20 PBS 1X solution (*v*/*v*). Afterwards, 150 μL of MSD-Gold read buffer was added to each well. This is a highly sensitive method, given the detection limits (LLD) of the analyzed cytokines: IL-10 = 0–3720 pg/mL (LLD = 0.14 pg/mL); IL-13 = 0–1920 pg/mL (LLD = 3.10 pg/mL); IL-8 = 0–2210 pg/mL (LLD = 0.15 pg/mL); MCP-1 = 0–6560 pg/mL (LLD = 0.74 pg/mL); and TNFα = 0–3650 pg/mL (LLD = 0.51 pg/mL).

The plate was read in the MESO QuickPlex SQ 120 MM model 1300 system (Meso Scale Diagnostics, LCC, Rockville, MD, USA), and the data were extracted by MSD discovery workbench analysis software. The cytokines were reported in pg/mL and then transformed into natural logarithms. Additionally, a BM inflammatory index (BM-II), based on anti- and pro-inflammatory cytokines, was calculated as follows:BM-II = (IL-10 + IL-13)/(IL-8 + MCP-1 + TNFα)

The scale typification of the individual cytokines and of the inflammatory index was used for the data analysis. In addition, the change (△) in the cytokine levels during the first month of lactation was calculated as follows:△cytokine = (level at day 28 − level at day 7)/level at day 7

### 2.7. Breast Milk Oxidative Status Analysis

The total antioxidant capacity of the samples was assessed by the 2,20-azino-bis- (3-ethylbenzothiazoline-6-sulfonic acid) radical cations (ABTS^∙+^) method, as previously reported [36]. Absorbance was measured at 734 nm in a plate reader (Cytation 5; BioTek; Winooski, VT, USA). Antioxidant capacity was calculated as mg Trolox equivalent (TE)/mL and then transformed into a natural logarithm.

Lipid peroxidation was assessed by malondialdehyde (MDA) and 4-hydroxy-*trans*-2-nonenal (HNE) level analysis by a kit (Lipid Peroxidation Assay kit KB-03-002, Bioquochem, Gijon, Spain) according to the manufacturer’s instructions, as previously reported [36]. Absorbance was measured at 586 nm in a plate reader (Synergy HT Multimode; BioTek; Winooski, VT, USA). MDA+HNE levels were expressed as μM and then transformed into a natural logarithm.

### 2.8. Statistical Analysis

Statistical analysis was performed using R software within the RStudio interface (version 2022.07.1+554, 2022, R Core Team, Vienna, Austria) using *rio, dplyr, compareGroups, ggpubr, devtools, stats, nlme, lme4*, and *ggplot2* packages.

The Kolmogorov-Smirnov test was used to determine whether the variables followed a known distribution. Quantitative variables were expressed as median and interquartile range [Q1; Q3] and qualitative variables as relative frequency and sample size (*n*). Quantitative variables were contrasted using an independent Wilcoxon test by day of lactation, and χ^2^ with Fisher correction was used for qualitative variables. Correlations were tested by Spearman’s rho (ρ) coefficient. A two-way ANOVA was used to test the differences between NAO, day of lactation, and the interaction effect (NAO * day). In the multivariate analysis, a linear mixed model was used, considering as a fixed effect the mDII and other significant factors reported in the univariate analysis. In addition, the woman was considered a random effect (RE). In all models, the coefficient (β) and the standard error (SE) were reported. The probability (*p*; *p*-value) to show significance was established at a value <5% in all analyses. In this study, imputation data techniques were not used.

## 3. Results

### 3.1. Sociodemographic Context and Obstetric and Neonatal Characteristics at Birth

Regarding sociodemographic data, 66.0% of the women were Spanish, with a median age of 34.0 years (min = 20.0; max = 44.0). 44.9% of the cohort had a university degree, and 55.1% had a high school degree. 71.4% of the women were employed.

With respect to obstetric parameters, the participants had a median of 2 previous pregnancies (min = 1; max = 3), and 94.0% of the women had a spontaneous gestation. Gestational diabetes was present in 24.5% of the cohort; 20.8% of the women had anemia; and preeclampsia was observed in 9.4% of the cohort. A C-section was performed on 29.1% of the women. 45.5% of the cohort had preterm births, and the median gestational age of the population was 37.3 [33.8; 39.0] completed weeks. No association between C-section and prematurity was detected (*χ*^2^ = 0.018; *p* = 0.89).

During the period of study, all infants had exclusive BM feeding, and 46.9% received antibiotic therapy. Males represent 43.6% of the cohort. Neonatal sex and prematurity were associated (*χ*^2^ = 6.284; *p* = 0.012). The median Apgar score was 10.0 [9.0; 10.0]. The neonatal antibiotherapy was not associated with C-section (*χ*^2^ = 0.821; *p* = 0.36) or prematurity (*χ*^2^ = 0.955; *p* = 0.33). NAO was diagnosed in 42.9% of the cohort, 22.2% for CLND, 12.7% for cardiac alterations, 7.9% for neurological alterations, 4.8% for IVH, 3.2% for LOS, and 1.6% for NEC. Premature birth was associated with NAO (*χ*^2^ = 8.380; *p* = 0.004). However, NAO was not associated with neonatal sex (*χ*^2^ = 0.09; *p* = 0.77).

### 3.2. Physiological Levels of BM Cytokines and Antioxidant Status during Lactation

To evaluate normal cytokine levels during lactation, BM from women with neonates diagnosed with a comorbidity were excluded. In BM from mothers with healthy neonates, IL-10, TNFα levels, and the inflammatory index of BM did not show significant changes during the first month of lactation. However, IL-13 levels increased while IL-8 and MCP-1 levels significantly decreased from day 7 to day 28 of lactation.

Total antioxidant capacity (by ABTS) and lipid peroxidation (by MDA+HNE) also decreased significantly during the first month of lactation (Table 1).

### 3.3. Influence of Sex, Gestational Age, and NAO on BM Cytokines and Oxidative Status

There were no differences in the levels of any of the studied cytokines between women with male or female infants, neither at day 7 nor at day 28 of lactation (Appendix A).

At day 7, the total BM antioxidant capacity of women with male neonates was significantly higher compared to women with female neonates, and no differences were detected at day 28. There were no sex differences in BM lipid peroxidation levels at day 7 or day 28 (Appendix A).

Gestational age was significantly and negatively correlated with IL-8 (ρ = −0.28; *p* = 0.019), MCP-1 (ρ = −0.25; *p* = 0.035), and TNFα (ρ = −0.26; *p* = 0.029). However, there was no correlation between gestational age and IL-10, IL-13, ABTS, or MDA+HNE.

The levels of BM cytokines and oxidative status during the first month of lactation were compared between women with healthy neonates and those with NAO. No statistical differences were detected in the BM levels of IL-10, IL-13, IL-8, TNFα, or the inflammatory index. However, the BM level of MCP-1 was higher in women with neonates with NAO. There was no interaction effect between the day of lactation and NAO on the levels of BM cytokines (Figure 2).

Regarding BM oxidative status, at day 7, ABTS tended to be higher in women with neonates with NAO. It was observed that during the first month of lactation, BM from women with infants diagnosed with NAO had a greater drop in antioxidant capacity, an interaction near statistical significance. The opposite was observed for BM lipid peroxidation levels in comparison with women with healthy neonates, with a significant interaction effect (Figure 3).

### 3.4. Influence of mDII, C-Section, and Obstetric Complications on BM Cytokines and Oxidative Status

The mDII was 0.97 [−0.71; 1.76] at day 7 and −0.84 [−2.41; 1.43] at day 28. No significant correlations were found between mDII and gestational age, BM cytokine levels, or oxidative status. At day 7, 26.2% of the women were categorized as low mDII and 37.7% as high mDII. At day 28, 24.6% of the women had a low mDII, 11.5% had a high mDII, and 85.0% maintained the level of diet inflammation. Furthermore, 15.0% of the cohort decreased their mDII during lactation, and there were no women who increased their mDII from day 7 to day 28.

Women who decreased their mDII during the first month of lactation had significantly higher levels of IL-10 and a trend towards lower levels of MCP-1 (Figure 4). However, we did not find a relationship between IL-13, IL-8, and TNFα levels and mDII category change from day 7 to day 28 of lactation.

At day 7, women with C-sections had significantly higher BM MCP-1 levels compared to vaginal labor, and no differences in other cytokines were detected. This difference was not observed at day 28 of lactation. BM total antioxidant capacity or lipid peroxidation levels were not different between women with vaginal labor or C-sections (Appendix A).

In women with obstetric complications, there were no significant differences in any individual BM cytokines except the BM inflammatory index, which was lower in women with obstetric complications at day 28. There were not significant differences in BM ABTS or MDA+HNE either at day 7 or 28 of lactation (Appendix A).

### 3.5. Association Factors Influencing BM Cytokines

Linear mixed-effect regression models were used to explore the effect of mDII on BM cytokines, considering women as a random effect and adjusted, in each case, by gestational age, day of lactation, and NAO. mDII did not have a direct effect on the level of any of the cytokines analyzed. Women with preterm labor had increased levels of IL-8 in their BM. The lactation period had a significant association with the levels of some BM cytokines, with IL-13 increasing during lactation while IL-8 and MCP-1 decreased. In addition, TNFα had the greatest explained variance as a random effect, indicating that this cytokine has high variability among women (Table 2).

## 4. Discussion

Cytokines are a relevant component of BM since they participate in the modulation of the immune response. Given the importance of BM for neonatal immune development, understanding the factors that influence the levels of pro- and anti-inflammatory cytokines present in the BM is relevant. In this work, we evaluated the influence of lactation period, neonatal (sex, prematurity, and NAO), and maternal factors (diet, C-section, and obstetric complications) on BM cytokines. We also evaluated the oxidative status and its relationship to the cytokine profile. The main findings from the present study are that, during the first month of lactation, BM progresses towards an anti-inflammatory cytokine profile. Among the neonatal factors, prematurity and NAO modulated BM cytokines with a higher proinflammatory profile. Regarding the influence of maternal factors, women who decreased mDII during lactation tended to improve the anti-inflammatory cytokine profile in BM. Instead, C-section increased BM MCP-1 levels, while obstetric complications did not have an influence on the cytokine profile. Regarding BM oxidative status, we found a quicker depletion of antioxidant capacity in women with NAO neonates. Linear mixed regression models evidenced that the most important factors modulating BM cytokines were lactation period and gestational age.

It has been reported that BM cytokines vary throughout the lactation period, with the highest levels found in colostrum [5,7]. However, not all cytokines follow the same trend. Our data showed that, during the first month of lactation, the pattern of cytokines evolved towards a more anti-inflammatory profile, increasing IL-13 and decreasing IL-8 and MCP-1. IL-8 decreased from day 7 to day 28, which agrees with the observations of other authors showing a slight decrease from the first to the third week of lactation [37]. Although IL-8 and MCP-1 are regarded as pro-inflammatory cytokines, a higher level may be beneficial at the beginning of lactation, when the neonatal bowel is more immature, contributing to gut mucosal defense and the development of the newborn’s immune system [37]. Higher levels of IL-10 were found in mature milk compared to colostrum [1]. We could not obtain colostrum for ethical reasons, keeping it for the neonate, and therefore we could not assess the changes in cytokines during the first days, but our data showed that levels of this anti-inflammatory cytokine remained stable from day 7 to day 28 of lactation. IL-10 plays an important role in the neonatal gastrointestinal tract, having an effect on the stabilization of B cells and decreasing the permeability of intestinal endothelial cells [2]. TNFα also remained stable during the first month of lactation. It has been described that this cytokine has a dual role since it can induce apoptosis in neonatal intestinal endothelial cells [2] but can also promote the production of anti-inflammatory cytokines [38]. It must be considered that the effects of TNFα are regulated via the soluble receptors I and II, also found in BM [39], which were not evaluated in the present study.

We analyzed the impact of premature birth on BM cytokine content since preterm infants have an underdeveloped immune system and the immune response mostly relies on non-specific innate immunity [14]. The placenta is a key endocrine organ that influences not only fetal growth but also immune development [40,41]. With premature delivery, placental signaling gets interrupted, and lactation becomes crucial for the continuous development of the neonatal immune system. The passage of maternal immunomodulators through the BM may help counteract neonatal deficiency. In this study, we observed a negative correlation between the levels of IL-8, MCP-1, and TNFα and gestational age, with IL-8 maintained in the adjusted models. This suggests that BM from preterm delivery has a pro-inflammatory pattern. In contrast with our study, there is evidence of a lower IL-8 content in preterm BM [37]. This difference could be related to the cytokine detection methods used or differences in the range of gestational ages studied. We used chemiluminescence MSD multiplex methodology, which enables us to quantify very small amounts of cytokines with high sensibility of detection, and in our cohort, there were very few extreme preterm neonates. We suggest that the higher levels of some cytokines we found in BM from women with preterm birth may be related to the fact that the mammary gland could be underdeveloped, with deficient tight junctions allowing for a larger passage of immunomodulatory peptides [2,42]. Among the cytokines analyzed, IL-8 is the smallest studied, with a molecular weight around 8 KDa [43], and therefore, this may be the reason for finding this peptide in larger quantities in women with premature delivery. We cannot rule out the role of maternal stress in the higher levels of pro-inflammatory cytokines found in preterm BM. Higher levels of IL-8 and TNFα have been found in women with stress [1], and mothers with preterm infants may be under stress conditions due to uncertainty about their health [44,45]. Despite their role in inflammation, premature infants could benefit from elevated levels of BM IL-8 and MCP-1 for gut development. These cytokines play a critical role in the movement of maternal immune cells to the BM [5] and, subsequently, across the neonatal bowel wall, contributing to mucosal defense and the development of the immune system of the newborn [37]. Even though higher levels of immunoreacting cytokines may help gut development, some neonates may display other comorbidities, such as respiratory pathologies, and high levels of chemoattractant cytokines could be detrimental, as evidenced in bronchopulmonary dysplasia [46,47]. Regarding how cytokines may pass through the stomach, it has to be noted that the pH in the neonatal stomach is higher, around 3–5 [48], which might allow more cytokines to reach the intestine without degradation and exert biological effects. Another possible explanation would be that cytokines are released embedded in exosomes, protected until they reach the intestine [3].

Cytokines are important for immune system regulation. However, an uncontrolled immune response results in excessive production, the so-called “cytokine storm”, with subsequent activation of immune cell populations [49]. The cytokine storm is also involved in the pathophysiology of neonatal diseases, particularly sepsis, although it is still not known what cascade points are beneficial to resolving infections and which are deleterious, leading to excessive inflammation [50]. Prematurity is a well-known risk factor for NAO, particularly in neonates with very low gestational ages [51]. We also found an association between prematurity and larger comorbidities in our cohort. Women with NAO neonates presented higher BM levels of the pro-inflammatory chemoattractant MCP-1. However, in the regression models, this association was lost. This could be related to modifications in the maternal immune response due to preterm delivery. However, we cannot discard the influence of the infant condition on maternal BM since it has been described that ongoing infection in neonates changes the BM macrophage profile [52].

Another common finding is the impact of fetal sex on preterm birth, with males at higher risk of prematurity [53], as observed in the present study. Prematurity is frequently associated with a worse outcome [53,54]. However, we did not find that male neonates had a worse NAO. This may be related to the fact that we had a few neonates of very low gestational ages, which are the most vulnerable. According to a Spanish data base, there is evidence that several neonatal morbidities (NEC, bronchopulmonary dysplasia, and LOS) are higher in very preterm males compared to females [55].

We also evaluated the influence of several maternal factors as modulators of BM cytokines, including obstetric complications, type of delivery, and nutrition. Our study provided evidence that MCP-1 was higher in women with C-sections, but no influence was observed with obstetric complications. Therefore, inflammatory maternal conditions related to gestation and delivery influence BM cytokines, similarly to chronic conditions such as asthma or celiac disease [10,11,56].

The influence of maternal nutrition on BM macronutrients has been extensively studied [57,58,59]. Evidence also suggests that maternal diet modulates BM bioactive compounds, such as lipids [60,61], antioxidants [62,63], and hormones [64]. In poor settings, the lack of quality proteins affects BM cytokine levels [19]. There is also evidence that maternal intake of n-3 LCPUFAs reduces BM pro-inflammatory cytokines and increases anti-inflammatory cytokines [1], while other studies do not show a modification with this supplementation [20,65]. However, a relationship between n-3 LCPUFA supplementation and plasma anti-inflammatory profiles has usually been demonstrated [66,67,68,69]. This discrepancy may be related to differences between cytokines in maternal plasma and in BM. We did not evaluate both matrices, and it would be desirable to perform such an analysis in future studies to determine the degree of passage of cytokines from plasma to BM. In our study, to evaluate the impact of diet on BM cytokines, we used mDII, which, to the best of our knowledge, has never been applied to the study of BM. In our study, although the maternal diet remained relatively stable during the first month of lactation, some women reduced the mDII, which was associated with increased BM IL-10 and a decrease in MCP-1. These results support the idea that the degree of inflammation in the maternal diet could have an influence on her BM cytokine levels. The importance of mDII on maternal health is evidenced in a study in Chinese breastfeeding women showing that a shift from a pro-inflammatory to an anti-inflammatory diet lowered the risk of postpartum depression [70].

Inflammation and oxidative stress are closely associated with many diseases, including those in the perinatal period. Obstetric complications, like gestational diabetes and preeclampsia, exhibit both an inflammatory and an oxidative stress component [24,25]. Furthermore, prematurity and NAO are also associated with oxidative disbalance due to deficiency in antioxidant systems, and BM provides a unique antioxidant profile beneficial for these neonates [71]. We analyzed the possible association between maternal or neonatal modulators of cytokines and BM oxidative status. We found an influence of sex on BM antioxidants, with higher levels in males. This may be related to the higher BM GSH levels previously found in male infants, since GSH is one of the most relevant antioxidants in BM [36]. We did not find a relationship between specific cytokines and oxidative status. However, in BM from children with NAO, which have higher levels of pro-inflammatory cytokines, we found evidence that the reduction in total antioxidant capacity of BM from days 7 to 28 was faster, while the lipid peroxidation levels showed the opposite trend. Since we did not assess maternal plasma oxidative status, we cannot be certain if our results are related to maternal factors associated with obstetric complications or prematurity. In our study, MCP-1 emerged as a common cytokine modulated by maternal and neonatal inflammatory conditions (C-section, NAO, and mDII). In many diseases with an inflammatory component, MCP-1 and oxidative stress have been associated, although the cause-and-effect relationship is not clearly established since MCP-1 can enhance ROS generation and phospholipid oxidation can also induce the release of MCP-1 [72]. This association deserves further studies, evaluating maternal plasma and BM oxidative status and cytokine levels.

### Strengths, Limitations, and Future Directions

The present work is a comprehensive study assessing relevant maternal and neonatal factors affecting BM cytokines. One of the strengths of our study is the use of mDII in the context of maternal nutrition since it is a global parameter. Our data support that maternal nutrition has a direct influence on BM cytokines and, therefore, diet modification can be a feasible way to promote an anti-inflammatory pattern, which may be beneficial for premature neonates with several comorbidities associated with inflammation, particularly respiratory diseases.

Regarding the limitations of the study, we did not evaluate the mechanisms through which maternal factors modulate BM cytokine levels, particularly the role of the microbiota. BM is a source of bioactive molecules, such as human milk oligosaccharides (HMO), which are able to increase anti-inflammatory cytokines through interaction with the microbiota [14]. In turn, microbiota modifications may modulate neonatal pathologies [73]. Microbiota may also be altered by antibiotics, which could explain the relationship between C-section and pro-inflammatory BM cytokines in our study. Another limitation is the lack of simultaneous determination of BM and maternal plasma cytokines and oxidative status, which is another area that deserves further study. A third aspect of future research is to obtain information on maternal dietary patterns prior to and during gestation and to consider other dimensions, such as sleep, psychological, and physical activity aspects, that may have effects on the bioactive components of BM.

## 5. Conclusions

The main factors affecting BM cytokines were the lactation period, gestational age, and diet. During the first month of lactation, the cytokine pattern evolves towards an anti-inflammatory profile. Preterm delivery and C-section have an influence on BM with higher levels of pro-inflammatory cytokines. Using mDII, we provide evidence of the influence of diet as a modulating factor to improve the anti-inflammatory profile in BM, supporting the relevance of maternal nutritional counseling, particularly important in women with preterm deliveries or C-sections.

## Figures and Tables

**Figure 1 antioxidants-12-00996-f001:**
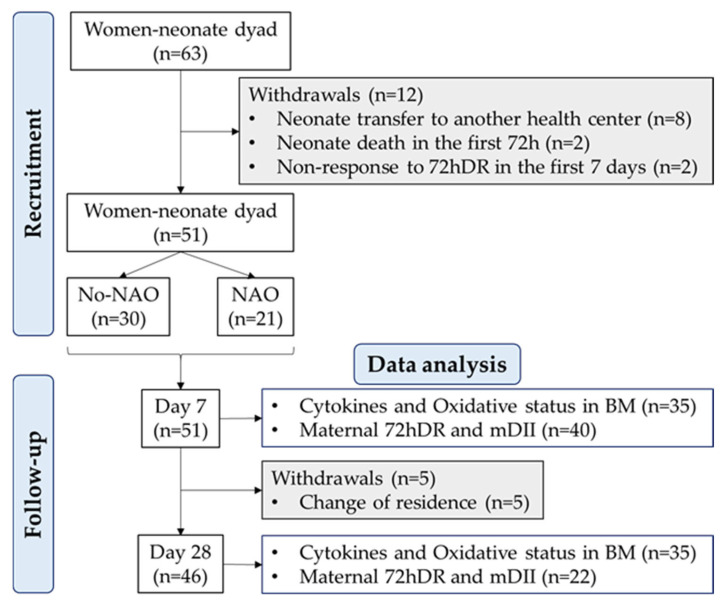
Flow chart of the study design. Neonatal adverse outcome (NAO); breast milk (BM); breast milk inflammatory index (BM-II); 72-h dietary recall (72hDR); maternal dietary inflammatory index (mDII); sample size (*n*).

**Figure 2 antioxidants-12-00996-f002:**
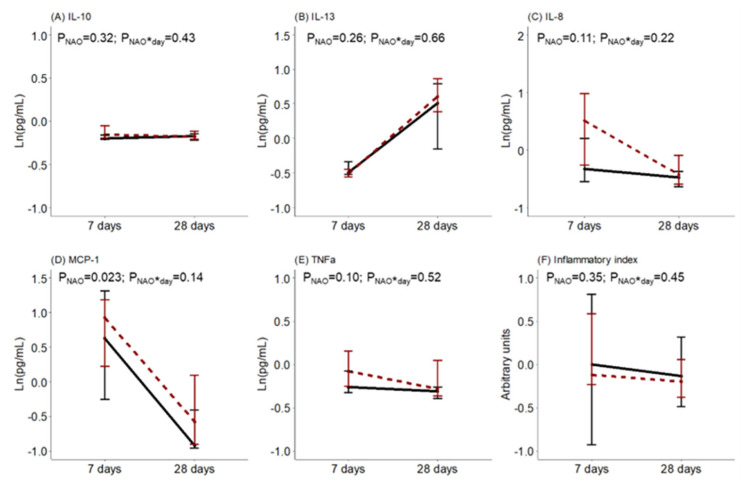
Breast milk (**A**) interleukin (IL)-10, (**B**) interleukin (IL)-13, (**C**) interleukin (IL)-8, (**D**) monocyte chemoattractant protein (MCP)-1, (**E**) tumor necrosis factor (TNF)α levels and (**F**) inflammatory index along lactation comparing women with healthy neonates (black continuous) or neonates with adverse outcomes (NAO, red broken lines). The data show the median, and the bar extensions show Q1 to Q3. The *p*-value (*p*) was extracted from a two-way ANOVA using NAO and the day of lactation interaction (*) as factors.

**Figure 3 antioxidants-12-00996-f003:**
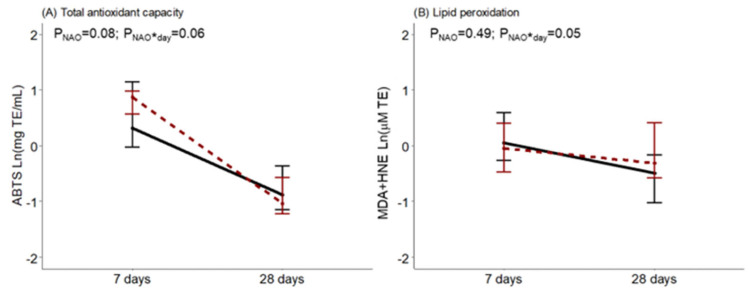
Breast milk total antioxidant capacity measured by the ABTS method (**A**) and lipid peroxidation measured by MDA+HNE (**B**) along lactation, comparing women with healthy neonates (black continuous) or neonates with adverse outcomes (NAO, red broken lines). The data show the median, and the bar extensions show Q1 to Q3. The *p*-value (*p*) was extracted from a two-way ANOVA using NAO and the day of lactation interaction (*) as factors.

**Figure 4 antioxidants-12-00996-f004:**
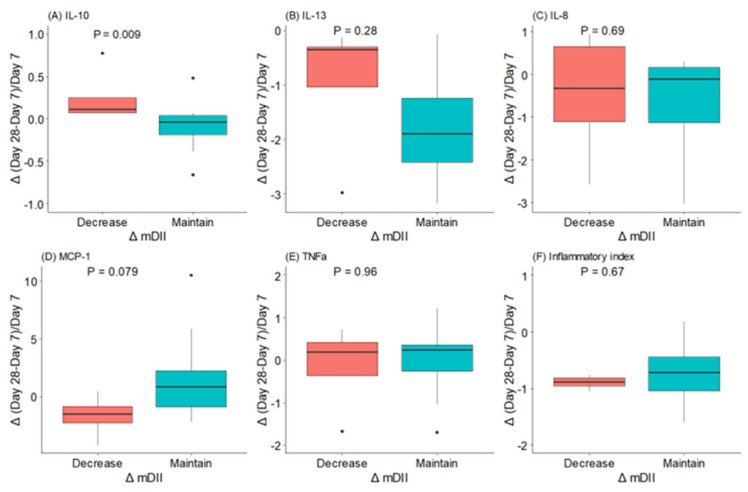
Changes (△) in maternal dietary inflammatory index (mDII) and in BM (**A**) interleukin (IL)-10, (**B**) interleukin (IL)-13, (**C**) interleukin (IL)-8, (**D**) monocyte chemoattractant protein (MCP)-1, (**E**) tumor necrosis factor (TNF)αv and (**F**) inflammatory index during the first month of lactation. The data show the median and interquartile range [Q1; Q3]. The *p*-value (*p*) was extracted from an independent Wilcox test. The dots mean the outliers, property of this type of analysis.

**Table 1 antioxidants-12-00996-t001:** Breast milk cytokine levels and oxidative status at day 7 and day 28 from women with neonates without adverse outcomes.

Cytokine and Oxidative Status	Day 7 (*n* = 20)	Day 28 (*n* = 18)	*p*
IL-10 Ln (pg/mL)	−0.20 [−0.21; −0.16]	−0.17 [−0.22; −0.15]	0.79
IL-13 Ln (pg/mL)	−0.49 [−0.52; −0.33]	0.51 [−0.15; 0.79]	<0.001
IL-8 Ln (pg/mL)	−0.32 [−0.54; 0.20]	−0.47 [−0.63; −0.37]	0.024
MCP-1 Ln (pg/mL)	0.63 [−0.25; 1.32]	−0.92 [−0.96; −0.41]	<0.001
TNFα Ln (pg/mL)	−0.26 [−0.32; −0.08]	−0.31 [−0.40; −0.26]	0.10
Inflammatory index	0.00 [−0.93; 0.81]	−0.13 [−0.49; 0.32]	0.87
ABTS Ln (mg TE/mL)	0.32 [−0.02; 1.14]	−0.88 [−1.14; −0.36]	<0.001
MDA+HNE Ln (μM)	0.05 [−0.26; 0.60]	−0.49 [−1.02; −0.17]	0.001

Data show the median and interquartile range [Q1; Q3] of cytokines after logarithmic (Ln) transformation and typification. The *p*-value (*p*) was extracted from an independent Wilcoxon test. ABTS: 2,20-azino-bis-3-ethylbenzothiazoline-6-sulfonic acid; TE: Trolox equivalent; MDA: malondialdehyde; HNE: 4-Hydroxy-*Trans*-2-Nonenal.

**Table 2 antioxidants-12-00996-t002:** Effect of mDII, gestational age, lactation period, and NAO on breast milk cytokines by linear mixed models.

	IL-10	IL-13	IL-8	MCP-1	TNFα	BM-II
mDII (Ref. “High”)	−0.03 ± 0.03 (*p* = 0.39)	−0.03 ± 0.12 (*p* = 0.12)	−0.06 ± 0.27 (*p* = 0.82)	−0.24 ± 0.25 (*p* = 0.34)	0.10 ± 0.21 (*p* = 0.63)	−3.89 ± 2.53 (*p* = 0.12)
Gestational age (Ref. Full-term infants)	-	-	0.55 ± 0.27 (*p* = 0.041)	0.15 ± 0.31 (*p* = 0.63)	0.60 ± 0.41 (*p* = 0.14)	-
Lactation period (Ref. day 7)	-	0.85 ± 0.12 (*p* < 0.001)	−0.64 ± 0.27 (*p* = 0.019)	−0.98 ± 0.22 (*p* < 0.001)	-	-
NAO (Ref. “no”)	-	-	-	0.17 ± 0.31 (*p* = 0.59)	-	-
Women as random effects	17.3%	8.8%	7.4%	35.4%	89.2%	4.2%
AIC/BIC	−57.5/−49.5	54.9/64.2	158.0/169.9	136.2/149.6	163.1/173.1	327.9/335.3

The data show the coefficient ± standard error (SE) and *p*-value (*p*). The models were built considering the references (Ref.): “high” category for maternal dietary inflammatory index (mDII); infants full-term; day 7 of lactation; and infants without neonatal adverse outcome (NAO). The explained inter-individual variance of the women is shown as the random effect. The Akaike Information Criteria (AIC) and the Bayesian Information Criteria (BIC) were used to fit the linear mixed models. Breast milk cytokines inflammatory index (BM-II).

## Data Availability

The data presented in this study are available in the article and Appendix A.

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
