# Peer review of "Maternal and Neonatal Factors Modulating Breast Milk Cytokines in the First Month of Lactation"

_antioxidants, 2023, doi:10.3390/antiox12050996_

Round 1

Reviewer 1 Report

The manuscript reviewed herein, "Maternal and Neonatal Factors Modulating the Breast Milk Cytokines in the First Month of Lactation" was designed to understand the shift in the concentration of breast milk cytokines during the first month of lactation.

The authors described well the study design, results and discussion.  There are a few points that authors should consider incorporating into the text.

1) Was all the healthy infant exclusively breast-fed?

2)To what extent were the premature infants receiving mother's milk vs donor's milk?

3) Was mother's or infant 's antibiotic intake recorded? For instance, women undergoing c-section usually receive antibiotic, which could potentially affect milk composition. Why didn't antibiotic intake be considered in the statistical analysis?

4) Was there an association between C-section and premature birth?

5) Line 453-455: "In the present study, we observed that a decrease in mDII along lactation was associated with increased IL-10 and a decrease in MCP-1 levels in BM, which supports that the degree of inflammation of maternal diet has an influence on her BM cytokine levels".  It appears to me that one cannot make this statement without the measurement of serum cytokines.

6)The inclusion criteria were: Maternal inclusion criteria were: ≥18 years old, good understanding of Spanish language, and non-diagnosed disease at the time of study. Mothers with specific dietary restrictions, such as those following a diet for competition sports, vegetarians, and vegans, were excluded from the study. However approximately 30% of the women had anemia or pre-eclampsia. Was there any association between these cases with milk cytokine levels.  I would think so.

Author Response

Response: Thank you for your time reviewing our manuscript. Your questions are responded below.

  1. Was all the healthy infant exclusively breast-fed?
  2. To what extent were the premature infants receiving mother's milk vs donor's milk?

Response: We have combined the answer to these questions. All healthy neonates were exclusively breastfed during the first 28 days of life and also preterm infants were fed on their mothers BM with no donated milk during the time of the study. This is now included in methods. 

  1. Was mother's or infant 's antibiotic intake recorded? For instance, women undergoing c-section usually receive antibiotic, which could potentially affect milk composition. Why didn't antibiotic intake be considered in the statistical analysis?

Response: This is an interesting point. Yes, women with C-section undergo antibiotic therapy during the 48-72h postpartum and we have now analyzed the influence of C-section. Some children also received antibiotics, which were recorded in the clinical data, and we have also assessed the influence. We have modified methods, results (new table S2) and discussion accordingly.

  1. Was there an association between C-section and premature birth?

Response: Although this is often a recurrent association, there was no association between prematurity and C-section in this cohort. This information has also been included (lines 280-281).

  1. Line 453-455: "In the present study, we observed that a decrease in mDII along lactation was associated with increased IL-10 and a decrease in MCP-1 levels in BM, which supports that the degree of inflammation of maternal diet has an influence on her BM cytokine levels". It appears to me that one cannot make this statement without the measurement of serum cytokines.

Response: We fully agree; therefore, we have modified the sentence and added the relevance to include the analysis of maternal plasma in the discussion and in the limitations of the study.

  1. The inclusion criteria were: Maternal inclusion criteria were: ≥18 years old, good understanding of Spanish language, and non-diagnosed disease at the time of study. Mothers with specific dietary restrictions, such as those following a diet for competition sports, vegetarians, and vegans, were excluded from the study. However approximately 30% of the women had anemia or pre-eclampsia. Was there any association between these cases with milk cytokine levels. I would think so.

Response: You raised a very interesting point that was now analyzed. Although expected, we did not find significant differences in breast milk cytokine levels between women with and without obstetric complications. This information was supplied in the text (new Table S3) and discussed.

Reviewer 2 Report

Manuscript presented by Ramiro-Cortijo et al. provides some data on maternal and neonatal factors modulating the breast milk cytokines during the first month of lactation. The concept is interesting. Nevertheless, the authors did not escape some ambiguities in the text. There are some areas that need to be addressed to improve the manuscript.

Please, find the comments below:

The authors should improve the English expressions by replacing them with accurate scientific English. The manuscript needs English proofreading overall, language revisions would be useful in order to make the manuscript more understandable.

Introduction:

An overview of existing current research requires some improvement overall. The are very few references from years 2020 to 2023, the rest of them are quite old.

Please, reconsider this aspect. 

Materials and Methods / Results:

I suggest adding detailed information on the process through a flow chart indicating the stages (e.g. recruitment, follow-up, data analysis), as well as withdrawals along the study, as it would give a better understanding of the process.

The study is in a small number of subjects to be of great significance. A larger study would have been better. I this regards, statistical section should include description on how the original sample size was calculated in order to detect significant statistical difference.

Considering that the present study has no control group. Is there any statistically significant differences between participants (mothers and neonates) for variables such as, lactation period, neonatal sex, or maternal diet. There should be some results (presented on a Table content) showing subjects baseline characteristics.

How did you test for a normal distribution in your variables?

It draws attention the fact that the authors did not include the dietary aspect during pregnancy among the data collection.

Even if the study is based during the first month of lactation, it should consider the diet during the pregnancy as well. There is the need to measure the regular food consumption of the women with 48 or 72 hour food recall together with a food frequency questionnaire, as it is directly related to maternal health, fetal development and especially the composition of the breastmilk.

In this sense, the lack of information about the gestational period is a relevant limitation. This is an important covariant that should be taken into account, or at least it should be explained in the discussion if the authors could not have the data.

The authors should acknowledge further limitations of the study.

The study seems a bit like a missed opportunity. The authors could have taken advantage of the study by doing a wider analysis to improve the quality of the results and the study overall. Conducting a wider analysis would add value to the work, e.g. inflammatory process, and particularly pro-inflammatory cytokines are closely related to oxidative stress and the modulation of the immune response.

Please, reconsider and explain these aspects.

Conclusions:

This section seems more like a missed opportunity. Authors are summarizing their results, which can be included in the conclusions, but they are not providing actual conclusions of their findings. e.g. What should be done in future studies?

Please, reconsider this aspect and explain it wider.

Author Response

Response: Thank you for your time reviewing our manuscript. Your queries were responded point-by-point and implemented in the text.

The authors should improve the English expressions by replacing them with accurate scientific English. The manuscript needs English proofreading overall, language revisions would be useful in order to make the manuscript more understandable.

Response: Thank you for this consideration. The manuscript has been now revised from language point of view to improve comprehension.

Introduction: An overview of existing current research requires some improvement overall. The are very few references from years 2020 to 2023, the rest of them are quite old. Please, reconsider this aspect.

Response: We have included some recent reviews on the subject and eliminated the old ones which were not relevant to the study.

Materials and Methods/Results: I suggest adding detailed information on the process through a flow chart indicating the stages (e.g. recruitment, follow-up, data analysis), as well as withdrawals along the study, as it would give a better understanding of the process.

Response: We agree with your comment. A flow chart with the recruitment, follow-up and analysis of variables was implemented as Figure 1.

The study is in a small number of subjects to be of great significance. A larger study would have been better. I this regards, statistical section should include description on how the original sample size was calculated in order to detect significant statistical difference.

Response: You are right. We have included the sample size calculation, which was based on the number of preterm births in Spain, since prematurity was a relevant aspect in our study. This is found in methods (lines 103-111).

Considering that the present study has no control group. Is there any statistically significant differences between participants (mothers and neonates) for variables such as, lactation period, neonatal sex, or maternal diet. There should be some results (presented on a Table content) showing subjects baseline characteristics.

Response: The baseline characteristics of the sample are described in the section 3.1.

How did you test for a normal distribution in your variables?

Response: The Kolmogorov-Smirnov test was used to determine whether the variables follow a known distribution. This information was implemented in the statistics section.

It draws attention the fact that the authors did not include the dietary aspect during pregnancy among the data collection.

Even if the study is based during the first month of lactation, it should consider the diet during the pregnancy as well. There is the need to measure the regular food consumption of the women with 48 or 72 hour food recall together with a food frequency questionnaire, as it is directly related to maternal health, fetal development and especially the composition of the breastmilk.

Response: We agree with this comment. The information of the nutritional pattern during gestation is important to determine the bioactive components of breast milk. However, it was not possible to obtain it, since our focus was on the woman during the lactation period, and we started the recruitment after birth. We have included this aspect as limitation of the study (last paragraph of the discussion).

In this sense, the lack of information about the gestational period is a relevant limitation. This is an important covariant that should be taken into account, or at least it should be explained in the discussion if the authors could not have the data. The authors should acknowledge further limitations of the study.

Response: We fully agree. Although we could not address lifestyle during pregnancy, we have now included analysis regarding the influence of pregnancy complications ad C-section which were recorded in this study (new Tables S2 and S3)

The study seems a bit like a missed opportunity. The authors could have taken advantage of the study by doing a wider analysis to improve the quality of the results and the study overall. Conducting a wider analysis would add value to the work, e.g. inflammatory process, and particularly pro-inflammatory cytokines are closely related to oxidative stress and the modulation of the immune response. Please, reconsider and explain these aspects.

Response: We fully agree on this point. Therefore, we have added data regarding oxidative status (total antioxidant capacity and oxidative damage to lipids) of breast milk and analyzed the possible relationship with cytokines and the analyzed maternal and neonatal factors. New information is included throughout the text, at the end of Table 1 and a new figure (Figure 3). Again, we agree that the best option would be to have data in plasma too, but this unfortunately was not possible.

Conclusions: This section seems more like a missed opportunity. Authors are summarizing their results, which can be included in the conclusions, but they are not providing actual conclusions of their findings. e.g. What should be done in future studies? Please, reconsider this aspect and explain it wider.

Response: We are aware of some limitations of the study, and we have included them in a section at the end of the discussion “strengths, limitations and future directions” which we hope can help to advance in the field of breast milk bioactive molecules.

Reviewer 3 Report

Thank you very much for the opportunity to review this very interesting paper. In particular, I found the study of the relationship between diet and inflammatory indices to be interesting and a study that will make a significant contribution not only in the field of obstetrics and gynecology but also in the field of nutrition. In order to make this excellent research more meaningful, I would like to make the following suggestions. If my suggestions are not academically appropriate, please disregard them.

    It would be good to describe whether pregnancy complications (e.g., gestational diabetes and gestational hypertension) do not affect BM, and the previous studies on this subject and the considerations in this study.

    I think the presents study need to consider the impact of lifestyle habits other than diet (exercise and sleep) on BM. If possible, it would be interesting if the DISCUSSION discusses its effect.

    It would be better to organize the paragraphs about the limitation and strength of this study at the end of the DISCUSSION.

Author Response

Response: Thank you for your time reviewing our manuscript. Your queries are  responded below, and the suggestions included in the text.

  • It would be good to describe whether pregnancy complications (e.g., gestational diabetes and gestational hypertension) do not affect BM, and the previous studies on this subject and the considerations in this study.

Response: This is a very interesting point. We have now analyzed the influence of obstetric complications in our study. There were no detected significant differences in the breast milk cytokine level. We also analyzed the impact of C-section, which was linked to higher levels of MCP-1. This information was supplied in the text and new tables as supplementary material (Tables S2 and S3).

  • I think the presents study need to consider the impact of lifestyle habits other than diet (exercise and sleep) on BM. If possible, it would be interesting if the DISCUSSION discusses its effect.

Response: You raised a very important point. Although we did not evaluate lifestyle in the present study, we have previously evidenced a poor adherence to physical activity in pregnant women and even worse during lactation period. We have now included this aspect in the limitations section.

  • It would be better to organize the paragraphs about the limitation and strength of this study at the end of the DISCUSSION.

Response: We fully agree. There are several limitations and possibilities of future work, including the abovementioned information about other lifestyle factors besides nutrition.  These aspects are now included in the section at the end of the discussion “strengths, limitations and future directions”.

Round 2

Reviewer 2 Report

The manuscript has been improved significantly. The authors have rearranged and clarified some information.

Authors have provided additional information to the introduction, which gives a global vision on the importance of the topic. They have also provided a flow chart describing the recruitment and the process along the study. They have added and clarified some aspects on the statistical section.

They have provided a practical point of view on their data by adding the section of Strengths, limitations, and they have explained what should be done in future studies.

The English language and style has been edited, which makes the manuscript more understandable.